# Social Media Usage and SME Firms' Sustainability: An Introspective Analysis from Ghana

**Emmanuel Bruce** [1,2,*], **Zhao Shurong** [2,3,*], **Sulemana Bankuoru Egala** [1,2], **John Amoah** [4], **Du Ying** [2,3], **Huang Rui** [2,3] **and Tai Lyu** [3,5]

[1] School of Management and Economics, University of Electronic Science and Technology of China, Chengdu 611731, China; sbegala@ubids.edu.gh

[2] Center for West Africa Studies, University of Electronic Science and Technology of China, Chengdu 611731, China; 202021160302@std.uestc.edu.cn (D.Y.); hr1999ht@163.com (H.R.)

[3] School of Public Affairs and Administration, University of Electronic Science and Technology of China, Chengdu 611731, China; cocolu@uestc.edu.cn

[4] Department of Business Administration, Faculty of Management and Economics, Tomas Bata University in Zlin, Mostni 5139, 76001 Zlin, Czech Republic; amoah@utb.cz or jamoah29@gmail.com

[5] School of International Education, University of Electronic Science and Technology of China, Chengdu 611371, China

\* Correspondence: kinbuki100@outlook.com (E.B.); shurz2015@ustc.edu.cn or shurz2015@163.com (Z.S.)

**Abstract:** Social media is gradually transforming diverse business ecosystems due to the limitless capabilities they offer. Given this, emerging businesses across the globe are leveraging this innovation to improve their operations. While the literature on the usage of social media by small and medium enterprises (SMEs) is still emerging, the outcomes from existing research have not been coherent. Amid this, limited empirical evidence has been adduced on the affordance of the technology for the SME ecosystem in developing economies, particularly Ghana. Following this, this study aims to fill this research gap by investigating the impact of social media usage on the long-term sustainability of SMEs, specifically in Ghana. Using empirical data from 424 respondents who are representatives of manufacturing SMEs in Ghana, using SmartPLS techniques, the study confirmed that, generally, social media usage does not only have a positive impact on SMEs but drastically drives their motivations towards resilience and sustainability. The results further revealed a positive and significant effect of social media usage value creation, business connections and opportunities on SMEs' sustainability. This study contributes to knowledge of social media usage and sustainability from a developing country's perspective. This study offers several implications for theory and practice.

**Keywords:** social media usage; SMEs; sustainability; manufacturing; Ghana



## 1. Introduction

Social media has exploded, infiltrating every aspect of individuals' and enterprises' socio-cultural life and forcing nations to technologically modernize. The rapid acceptance and use of social media has benefited small and medium businesses (SMEs), considerably assisting them in overcoming a long-standing problem of efficiently marketing their goods and services. As a result, by increasing the adoption of social media and taking the opportunities it provides, SMEs can achieve business sustainable growth. In recent times, the acceptance and usage of social media has influenced modern-day commercial activities [1]. Kaplan and Haenlein [2] defined social media as *a group of Internet-based applications that build on the ideological and technological foundations of Web 2.0, and allow the creation and exchange of user-generated content"*. Moreover, the emergence of social media has improved communication and the sharing of ideas with others through various social media channels [3]. Today, the pandemic (Corona Virus Disease, COVID-19), has had severe consequences for businesses [4]. The fear of physical contact caused by the pandemic has

moved customers to resort to social media to seek information and make purchases [5]. With this in mind, social media is seen as a new innovative tool that would only climb sharply in the dynamic business environment. Generally, social media is considered to be an innovative tool for business performance [6–8]. Moreover, the integration of comprehensive social media marketing into business processes has yielded positive results [9–11].

Globally, small and medium enterprises' (SMEs) contribution is pivotal to economic growth [12]. According to [13], SMEs consist of almost 90% of businesses worldwide. Principally, SMEs contribute significantly to economic development through new product development and job creation [8,12]. Moreover, SMEs also provide a ground for entrepreneurs to enhance their creativity and invent something new. On account of that, a developing country needs a vibrant and vigorous SME sector for economic development, alleviating socioeconomic inequalities, and overall contribution to the Gross Domestic Product [14]. Recently, businesses are more likely to adopt new technology for firms performance. For instance, Eurostat [15] reported that multinational companies in the European Union (EU) are creating values and interacting with both potential and loyal customers through social media. On the contrary, Ref. [16] found that only 30% of SMEs have taken advantage of this innovative tool for business performance. As a result, social media adoption can be vital for SMEs, which lack marketing and advertising resources [17]. More instructively, Qalati et al. [18] suggest the adoption and usage of social media for SMEs' sustainable performance. With this assertion, SMEs need to invest in new technologies to build customer relationships that could enhance productivity and performance [19]. Moreover, Ref. [20] noted that only a small proportion of SMEs are effective in attaining unprecedented performance and sustainable growth. This is quite telling given the fact that SMEs across the globe are fast evolving amid changing business trends mediated by technology to drive their sustainable growth. As a result, it is very necessary to consider social media as a new technological tool and its effect on SMEs' sustainability.

The literature has established the association of social media usage with a firm's performance in the context of SMEs in developing countries [3,4,8,19]. Social media is pivotal to SMEs' development of communication strategies [21]. Moreover, social media provides channels to effectively communicate with customers through customized products and innovative experiences. However, challenges exist when adopting social media to achieve sustainability in the context of SMEs from developing countries' perspectives [4]. Moreover, social media usage in the SME context and its sustainability remains unclear in developing countries [12]. Hitherto, only a few studies have investigated social media as an innovative tool for SMEs' sustainable performance [22–24]. Other scholars argue that SMEs' social media usage for long-term sustainability has been a challenge [3,4]. Based on the limited previous literature, specifically in the SMEs context, this paper seeks to advance a research gap by examining the effect of social media usage on SMEs' sustainability from the developing country's perspective, particularly Ghana. The study adds to the knowledge of social media's impact on SMEs' sustainability, particularly in developing economies. Additionally, the study contributes to the discourse on the relationship between social media usage and SMEs' sustainable performance.

## 2. Theoretical Background: Uses and Gratification Theory (U&G)

It is evidenced that businesses will see substantial returns if they are successful in establishing connections with customers through social media [19,23]. Despite the significance of social media, theoretical understanding regarding how SMEs use social media and the reasons why they do so is still lagging. According to the theory of usage and gratification (U&G), individuals deliberately seek out particular media to satisfy their requirements [25]. The U&G theory examines the factors, both psychological and social, that lead individuals to select particular content and media channels, as well as the resulting attitudes and behaviors [26]. The U&G theory has its origins in communications literature and plays an important role in the process of enhancing the scales and measurement tools available to social media marketers. The fundamental idea behind the uses and gratifications theory is

that people look for forms of entertainment that cater to their specific requirements and ultimately satisfy them [27]. Even though the U&G theory has a lot of bearing on the topic of social media, it has not received a lot of attention in the research that has been performed on marketing and social media.

Extant studies have applied the U&G theory in similar studies on social media usage. Whiting and Williams [27], for instance, leveraged the U&G theory to explore why consumers use social media. Similarly, Kamboj [28] used the theory to investigate the effect of gratifications gained by customers on social media on brand trust and commitment. Sun et al. [29] explored how employees of enterprises use social media to engage their audience. In the current study, we draw on the U&G theory to investigate how social media usage promotes SME firms' sustainability. First, we synthesize the theory by elucidating the constructs from the theory to understand why SMEs in Ghana leverage social media to engender the performance of their firms towards sustainability. Second, the fundamental idea underlying the U&G theory, according to Whiting and Williams [27], is that individuals will seek out, among various competing forms of media, the one that satisfies their requirements and ultimately satisfies their desires. Given the above, it is obvious that social media offers several capabilities to SMEs in promoting their marketing mix. Hence, we argue that drawing on the U&G theory as a theoretical lens is significant.

## 3. Literature Review

### 3.1. Social Media

According to [12], social media enables firms to increase their business growth. Kaplan and Haenlein [2] advanced that social media provides networks through which people can share information and build networks. Social media has been considered an effective marketing tool compared to traditional media such as television, radio and newspaper [30]. A study by [31] also identified connectedness, transformation and relationship building as the three underpinnings of social media. In this sense, social media serves as a channel for generating new ideas, creating value, and effective marketing [32]. Social media can help SMEs to innovate and compete in this current business environment [33]. Moreover, SMEs can use social media firms to establish creativeness to control the current market situations [34]. A study conducted by [4] maintained that SMEs can use social media platforms to attract customers and, in addition, obtain adequate information about the product/service before making a purchase decision. Furthermore, Ref. [35] added that businesses connect with a larger audience, potential suppliers and other businesses via social media platforms. Social media can also help SMEs to receive feedback from customers and build customer relationships. Other scholars have confirmed a significant positive influence of social media on SMEs' sustainable performance from the developing countries' perspective [14,36]. Thus, this study hypothesizes that following:

**Hypothesis 1 (H1).** *Social media as an advertising tool would positively affect SMEs' sustainability.*

### 3.2. Information Channel

Prior studies have extensively investigated the usage of social media as an information channel for SME development [24,37]. Social media can be adopted as an information channel for SMEs' effective marketing [38]. In the context of SMEs, social media has proven to be a platform that can be utilized by SMEs to disseminate information to customers and the masses [4]. Preliminary work by [3], established that SMEs use social media platforms to interact, socialize, communicate and collaborate with businesses and potential investors. For instance, a study conducted by [4] investigated social media as a promotional tool for SMEs' development. The study used the financial sector in Ghana as a case study and revealed that social media improves SMEs' communication strategies. Similarly, Ref. [39] examined social media as a communication channel among SMEs in developing countries and confirmed that social media has a positive effect on SMEs' sustainable growth. Additionally, Ref. [31] also found a significant relationship between social media and

business performance among Palestine SMEs. Furthermore, Ref. [40] recently evidenced that SMEs' social media usage positively influences customer attraction, interactivity and customer relationships. Considering the above evidence, we hypothesized that:

**Hypothesis 2 (H2)**. *Social media as an information channel would positively affect SMEs' sustainability.*

### 3.3. Value Creation

The interactive feature of social media has enabled SME firms to evaluate and probe the external environment and other factors for organizational performance [36]. Several studies have confirmed the association between social media usage and value creation [41,42]. Ahmad et al. [14] observed that social media positively drives value creation. According to [29,43] social media positively influences value creation by building networks with both existing and potential customers through information flow. Similarly, Ref. [42] collected data from social media users online and confirmed a positive significance between social media usage and value creation. Furthermore, Ref. [38] submits that SMEs can create value through customer-driven sales. A paper published by [44] also observed a positive influence of social media usage on SMEs' value creation. The findings in [45] further argue that organizational learning mediates between social media marketing and value creation. Their study confirmed that social media usage positively affects market shares through organizational learning. However, Ref. [46] found an insignificant influence of social media on the value creation of SMEs in Spain. Finally, a study conducted by [8] proposed SMEs should emphasize their pursuit to compete and innovate by presenting goods and services along with social media to add value to the firm. Based on this review, we state the hypothesis that:

**Hypothesis 3 (H3)**. *Value creation through social media usage would positively affect SMEs' sustainability.*

### 3.4. Business Connection and Opportunities

The introduction and development of social media have expanded and created opportunities for SMEs to link between customers and potential partners [9]. According to [47], the emergence of social media has enabled SMEs to create networks with suppliers and also make rapid communications both locally and internationally. From the SMEs' perspective, Ref. [3] found that social media usage creates networks that lead to the superior performance of SMEs in developing countries. Moreover, the usage of social media by SMEs drives them to compete with larger firms, hence gaining a competitive advantage [48,49]. Furthermore, managers of SMEs can share knowledge, create customer relationships and provide better customer service through social media platforms [9,50,51]. A study conducted by [52] maintained that social media has a positive effect on creating SMEs' business connections, particularly with customers and other stakeholders. Consequently, a study by [53] affirmed that creating business connections with stakeholders leads to customer loyalty and firm reputation. Recent work by [54] on social media usage on SMEs' performance observed a positive association between social media usage and SMEs' business performance. Moreover, the findings of [55] remarked that social media allow SMEs to monitor the consumer decision-making process and to make informed decisions. Henceforth, based on the literature, we propose that:

**Hypothesis 4 (H4)**. *Business connections and opportunities through social media usage would positively affect SMEs' sustainability.*

### 3.5. Sustainability

According to [56], sustainability comprises well-balanced financial resources and environmental, technological and social-economic objectives. The authors added that

helping to serve, save and maintain the community, environment and economy is crucial to SMEs' sustainability. The study by [57] suggested that SMEs should use technologies to keep pace with fast-changing market situations. In doing so, Refs. [12,55] revealed that SMEs need to acquire innovative tools such as social media marketing to build long-lasting relationships with their stakeholders through collaborative information. Thus, SMEs constantly improve communication through social media, which subsequently leads to business success [58,59]. Very few studies have found a significant relationship between social media usage and SMEs' sustainability from a developing county's perspective [22,24]. For instance, Ref. [36] established that social media usage improves SMEs' performance and reduces resource shortages. Additionally, Ref. [60] also observed that social media usage reduces side effects on the SME environment within which they operate. This implies that the effectiveness of social media platforms has made it possible for SMEs to scan, monitor and evaluate the external environment and create new business opportunities, which are significant for organizational sustainability [61]. Moreover, Ref. [62] revealed a positive influence of social media usage on SMEs' value creation and sustainability. Empirical observations by [3] on social media usage on SMEs' sustainability gathered a sample from SMEs in Ghana. The study findings proved social media networks have enabled SMEs to achieve other organizational objectives such as customer engagement, value creation, network building and business sustainability. Based on the literature above, we propose that:

**Hypothesis 5 (H5)**. *Social media usage would positively affect the sustainability of small and medium enterprises (SMEs).*

## 4. Methodology

### 4.1. Sample, Data Collection and Analytic Techniques

This study largely focused on SMEs in the manufacturing sector for its data collection. The researchers adopted a non-randomized sampling technique, particularly convenience sampling, to select the study respondents/participants. The structured questionnaire was divided into two sections. Section A of the questionnaire contains the demographics of the respondents, while the second part contains questions on the study constructs. In all, 35 questions were answered by the respondents. Again, a structured questionnaire was developed and distributed among SME firms in the manufacturing sector of Ghana. The adoption of convenience sampling in this study is deemed to be appropriate since recent articles on the confinement of SMEs have also used it, based on the participants' availability and eagerness to produce the required information needed to execute this study objective [4,63] and, finally, based on the following advantages: quick data collection, easy to conduct research, low cost, fewer rules to follow and reduction in spent time. A total of 500 structured questionnaires were distributed, of which 424 represented (84.80%) were appropriate to be used in the data processing and analysis. This means that 76 questionnaires had some anomalies that made them unfit to be used in the data processing and analysis. Additionally, Ref. [64] maintained that quantitative research must obtain at least 300 respondents for data processing and analysis. In this regard, the current study meets such a requirement with 424 respondents after eliminating the anomalies and invalid responses. Relatedly, the researchers adhered to the adoption of both online and offline data collection processes. Permission was first sought from the selected organization through emails, WhatsApp messages and letters, among others, on the participation in the data collection processes. However, the offline data collection was subsequently performed, taking into consideration the various restrictions enacted by the firms as a result of the COVID-19 pandemic, while the online data collection took the form of sharing the Google form link. The structured questionnaire was, therefore, answered by both management and employees of the selected SMEs.

The study adopted management and employees in answering the questionnaire based on the substantial information in their possession, and again, to obtain some level of

concrete evidence that would be beneficial to both theory and practice. To add more, the questionnaire was administered to only SMEs in the manufacturing sector of the country who primarily used social media platforms to drive their business growth towards its sustenance. It is quite important to highlight that the cross-sectional research design was deployed in the data collection process since the data would be processed and analyzed once as compared with the longitudinal research design approach [65].

Before embarking on the main data collection, a pilot study was conducted to test the constructs' reliability and validity through the expected values of Cronbach alpha, of which 50 respondents were used in the exercise. Moreover, the researchers used five months in the data collection process, particularly from January to May 2022. Each respondent had a maximum of ten minutes to answer the stated questions. The researchers advised the respondents not to indicate their details/particulars. This is to ensure a high ethical standard of research, as revealed by [3]. To finalize the methodology, a PLS-SEM partial least squares and structural equation modeling), specifically, the ADANCO 2.2.1 version of the software, was used to process the study data after taking out the incomplete, duplications and indifferences in some of the answered questionnaires, the conceptual framework and, lastly, the study hypotheses. To elucidate the reason for using PLS-SEM over AMOS and Lisrel is that Smart PLS can process data from both the formative and reflective SEM models. The Table 1 below depicts the details of the respondents' information used in this current study (see Appendix B for article questionnaire and measurement).

**Table 1.** Background Information of Respondents.

| Details | Frequency | Percentage (%) |
|---|---|---|
| Gender | | |
| Male | 301 | 70.90 |
| Female | 123 | 29.10 |
| Age | | |
| 18–25 years | 130 | 30.70 |
| 26–35 years | 176 | 41.40 |
| 36–45 years | 75 | 17.70 |
| 45–55 years | 26 | 6.20 |
| Above 55 years | 17 | 4.00 |
| Educational Level | | |
| SSSCE | 126 | 29.80 |
| Diploma/HND/Bachelors | 184 | 43.30 |
| Masters/Post Graduate Diploma | 83 | 19.60 |
| Others | | |
| Organizational Size | 31 | 7.30 |
| Medium 1–10 employees | 170 | 40.00 |
| Small (11–50 employees) | 126 | 29.80 |
| Medium (51–100 employees) | 65 | 15.40 |
| Large (above 100) | | |
| Years Served | 63 | 14.80 |
| 1–5 years | 235 | 55.30 |
| 6–10 years | 110 | 26.00 |
| 11–15 years | 43 | 10.20 |
| Above 15 years | | |
| SM Devices Used | 36 | 8.50 |
| Mobile Device Only | 31 | 7.30 |
| PC/Computer | 93 | 22.00 |
| Both | | |
| Number of times SM used by the Company | 300 | 70.70 |
| Daily | 206 | 48.50 |
| Once a week | 115 | 27.20 |

**Table 1.** *Cont.*

| Details | | Frequency | Percentage (%) |
|---|---|---|---|
| | More than once a week | 55 | 13.00 |
| | Once a month | | |
| | SM Platforms Used | 48 | 11.30 |
| | Facebook | 126 | 29.60 |
| | Twitter | 33 | 7.70 |
| | Instagram | 41 | 9.70 |
| | LinkedIn | 32 | 7.60 |
| | All the above | 192 | 45.40 |
| Sample Size Used | | 424 | 100 |

Source: Field data (January 2022–May 2022), retrieved from Google form.

### 4.2. Measurement of the Constructs

The researchers took inspiration from previous studies in ascertaining the constructs' validity. The study constructs were, therefore, taken as sustainability [4,22], business connection and opportunities [54,61], value creation [4,36], information channel [58,66] and, the predictor construct, social media usage [3,37,66]. The measurement of the study constructs was performed through the usage of the five-point Likert scale, Completely Disagree (1), Disagree (2), Neutral (3), Agree (4) and Completely Agree (5).

### 4.3. Test of Common Method Variance (CMV)

Since the current study draws its data independently, the possibility of common method variance is very high. Additionally, the participants used in this study were assured of protecting their data as confidential and informed that there was no wrong or right answer to every question they answered in this survey. Per the research of [67], the presence of Common Method Bias (CMB) was ascertained, which informed the researchers to subsequently design the questionnaire with the description on the title page and to treat respondents or participants with utmost confidence. To be more precise, the questionnaire was developed such that the respondents or participants could opt out as and when they wanted to do so. Above all, the researchers performed a multicollinearity test concerning VIF (variance inflation factor) to reveal the existence of Common Method Bias (CMV). The results of a post-hoc evaluation reveal that CMV has a minimal existence based on VIFs (see Table 2) where the thresholds are less than ten (10), as revealed by [68–70]. Finally, on the issues of CMB in this survey it is regarded to be minimal. Therefore, the CMB is of less concern.

**Table 2.** Test of construct items, loading and variance inflation factor (VIF).

| Construct | Indicator | Loading | VIF |
|---|---|---|---|
| INFORMATION CHANNEL | IC1 | 0.8561 | 2.7978 |
| | IC2 | 0.8936 | 3.6393 |
| | IC3 | 0.9086 | 3.9216 |
| | IC4 | 0.9274 | 6.0051 |
| | IC5 | 0.9105 | 5.0577 |
| VALUE CREATION | VC1 | 0.9127 | 3.9188 |
| | VC2 | 0.9062 | 3.9983 |
| | VC3 | 0.9245 | 4.7885 |
| | VC4 | 0.9331 | 5.1645 |
| BUS. CON & OPP | BCP1 | 0.9245 | 4.7497 |
| | BCP2 | 0.9203 | 4.7734 |
| | BCP3 | 0.9387 | 5.6589 |
| | BCP4 | 0.9456 | 6.9488 |
| | BCP5 | 0.9381 | 6.1640 |

**Table 2.** *Cont.*

| Construct | Indicator | Loading | VIF |
|---|---|---|---|
| SOCIAL MEDIA | SM1 | 0.8175 | 2.0103 |
| | SM2 | 0.9129 | 3.4578 |
| | SM3 | 0.9101 | 3.4562 |
| | SM4 | 0.9231 | 3.8704 |
| SUSTAINABILITY | SS1 | 0.9225 | 4.5068 |
| | SS2 | 0.9396 | 5.5885 |
| | SS3 | 0.9348 | 5.3692 |
| | SS4 | 0.9342 | 5.2511 |
| | SS5 | 0.9331 | 5.1000 |

Source: Authors' processing from ADANCO 2.0 version.

## 5. Empirical Findings and Results

### 5.1. Model Measurement

The constructs' reliability and validity were vigorously measured through Dijkstra–Henseler's Rho with Cronbach alpha coefficients, as the researchers were motivated by the PLS-SEM application literature of scholarly works [64,71]. Since the values of the coefficients are all above 0.5 thresholds (see Table 3 below), it indicates the strongest of the coefficients of the constructs as established by [67,71]. The adoption of the ADANCO 2.0 version was used in assessing the psychometric properties concerning the underlying items of the research constructs. Again, the composite reliability of the constructs, as shown in Table 3, recorded 0.7 and 0.8 as the minimum and maximum thresholds concerning Jöreskog's Rho (pc) and Dijkstra–Henseler's Rho (pA), which fulfills the basic requirements. With regards to Dijkstra–Henseler's Rho (pA), 0.9215 and 0.9635 were recorded as the coefficients' construct reliability, and, finally, a minimum threshold of 0.5 was recorded regarding the average variance extracted (AVE), which stands for convergent validity, as revealed in Table 3. As revealed by [67], all the factor loadings of the constructs were importantly assessed and loaded to their respective positions, which met the requirement of 0.6, showing how good the indicators are. From the below table, the coefficients of the respective constructs were all above 0.6, showing 0.8978 as the minimum and 0.9456 as the maximum loading. The details of the research constructs with their corresponding loadings are all shown in Table 3 below. Further, the issue of multicollinearity was of great concern to the researchers and was detected with the help of the common method variance (CMV) through the scale measurements of variance inflation factor (VIF). As per the works of [3,72], CMV is not an issue since the VIF is less than 5 against a maximum threshold of 10. The factor loadings of the research constructs are, therefore, shown in Table 3 below.

**Table 3.** Test of validity and reliability of the research construct.

| Constructs/Indicators | Dijkstra–Henseler's Rho (ρA) | Jöreskog's Rho (ρc) | Cronbach's Alpha (α) | The Average Variance Extracted (Ave) |
|---|---|---|---|---|
| INFORMATION CHANNEL | 0.9431 | 0.9549 | 0.9409 | 0.8092 |
| VALUE CREATION | 0.9516 | 0.9625 | 0.9513 | 0.8371 |
| BUS. CON & OPP | 0.9635 | 0.9713 | 0.9631 | 0.8714 |
| SOCIAL MEDIA | 0.9215 | 0.9395 | 0.9137 | 0.7955 |
| SUSTAINABILITY | 0.9627 | 0.9710 | 0.9627 | 0.8702 |

Source: Authors processing from ADANCO 2.0 version.

Notwithstanding, Henseler et al. [73] inspired the researchers to evaluate the existence of the discriminant validity of the latent variables through Fornell–Larcker's criterion analysis [74]. As established by experts, such as [71,73], all the values in the diagonal form (bold) exceed the minimum requirement of greater than 0.5, which reveals the average variance extracted (AVE) of the measured constructs (see Table 4 below). Fornell–Larcker's criterion of the discriminant validity shows the basic and stringent assumptions of the

research constructs were established once each construct of AVE had higher coefficients (both column and row position) than the other constructs.

**Table 4.** Test of Discriminant Validity—Fornell–Larcker criterion.

| | | | | | |
|---|---|---|---|---|---|
| 1 = Information Channel | **0.8092** | | | | |
| 2 = Value Creation | 0.5649 | **0.8371** | | | |
| 3 = Business Connections and Opp. | 0.4600 | 0.4985 | **0.8714** | | |
| 4 = Social media | 0.5220 | 0.3945 | 0.3137 | **0.7955** | |
| 5 = Sustainability | 0.3835 | 0.5104 | 0.5284 | 0.2966 | **0.8702** |

Note: the *diagonal (in bold)* is the average variance extracted (AVE) Sources: Author's processing from ADANCO 2.0 version.

## 5.2. Structural Modeling-Path Analysis

The researchers saw the essence of the path analysis, otherwise called structural modeling, in this current study, which concerns the model fit. The significance of this analysis is to reveal the causal effect of the research constructs. Therefore, the findings of the study show that social media (SM) has a potential significant effect or impact on the current research constructs, such as information channel (IC), value creation (VC), business connections and opportunities (BCO) and sustainability (SS). Table 5 below, therefore, shows the regression coefficients of Beta ($\beta$), significant values; T-values >1.96 (or *p*-values < 0.05) concerning the research model. Additionally, the predictive power that concerns the research model of the values determination of the regression model was evaluated. Hence, the $R^2$ values of the independent variables are information channel (IC) 52%, value creation (VC) 39% and business connections and opportunities (BCO) 31%, and the dependent variable (sustainability, SS) has an $R^2$ value of 60%, as seen in the table and Figure A1 (Appendix A).

**Table 5.** Hypothetical path coefficient.

| Relationship | Beta ($\beta$) | Standard Bootstrap Results | | | | | Empirical Remarks |
|---|---|---|---|---|---|---|---|
| | | Mean Value | SD Error | t-Value | Effect Size (Cohen's f$^2$) | *p*-Value | |
| H1: SS -> SS | 0.0075 | 0.0072 | 0.0758 | 4.5172 | 0.2217 | 0.0001 | **Agreed** |
| H2: IC -> SS | 0.0011 | 0.0067 | 0.0805 | 0.0137 | 0.0000 | 0.4945 | **Not Agreed** |
| H3: VC -> SS | 0.3654 | 0.3678 | 0.0787 | 4.6443 | 0.1227 | 0.000 | **Agreed** |
| H4: BCO -> SS | 0.4257 | 0.4280 | 0.0634 | 6.7126 | 0.2100 | 0.1282 | **Agreed** |
| H5: SM -> SS | 0.5446 | 0.0643 | 0.0668 | 1.1358 | 0.0068 | 0.000 | **Agreed** |
| *Independent variable:* | Coefficient of determination ($R^2$) | | | Adjusted $R^2$ | | | |
| Information channel | 0.5220 | | | 0.0.5209 | | | |
| Value creation | 0.3945 | | | 0.3930 | | | |
| Business connections and opportunities | 0.3137 | | | 0.3120 | | | |
| *Dependent variable* | | | | | | | |
| Sustainability | 0.6125 | | | 0.6088 | | | |

Source: Author's processing from ADANCO 2.2.1 software.

## 6. Discussions and Implications

Research evidence has helped the significance of social media in supplementing the convergence of technologies used by SMEs with substantial benefits towards their sustainability [3,36,75,76]. While the capabilities of social media have transcended from the traditional information medium, it has become a strategic tool for transforming businesses and organizations [4,77–79]. This trend has given businesses a compelling competitive urge, making them more sustainable than before. Social media is a key enabler in consumer marketing mix strategy for connecting directly with the targeted audience while maintaining a constant interaction with customers to retain them. Realizing these, this study set out to investigate how social media could be leveraged to drive and sustain the growth of SMEs.

Thus, we specifically examined the effect of social media on the sustainable development of SMEs in Ghana. The study hypothesized that social media has a positive effect on the sustainability of SMEs. As affirmed by [22,24], the capabilities of social media have made SMEs creative by exploring market trends and tailoring marketing campaigns toward their target customers, increasing their share and subsequent profitability. In effect, social media has an extremely positive impact on the profitability of SMEs, thereby making them competitive [3,4,80–82]. Supporting the existing research [24,36,46], this study's results also show that social media plays a crucial role in achieving SMEs' sustainability.

Again, the study hypothesized that the use of social media as a channel of information would positively impact the sustainability of SMEs. This proposition was, however, not supported, although extant studies have documented how social media is drastically being harnessed to support the information communication needs of businesses [3,4]. In addition, the current study establishes that information channels have an indirect effect on SMEs' sustainability. Yet the affordability of social media as an effective marketing information channel for businesses cannot be discounted [83]. Even though this assertion was not supported, the affordability of social media as an effective marketing information channel for businesses cannot be discounted [24,29,84]. Corroborating the above assertion, Ref. [34] intimated that social media helps in the continuous interaction between businesses and their customers. Social media platforms such as Twitter and Facebook are among the effective information channels often deployed by businesses for customer engagement [7,26,85]. Specifically, Ref. [85] stated that social media platforms significantly impact firms' operational activities since it spurs brand development and improve customer traction.

It also emerged from the findings that social media offers SMEs the opportunity to create value for their sustainability. Thus, the result shows a positive correlation between social media usage and value creation. This proposition was supported and confirmed by extant studies [36,37,41]. The findings from [41], for instance, conclude that, besides the ability to scan the firm's environment using user-generated comments on social media, businesses leverage their social platforms to create value through the generation of new business opportunities and build a brand reputation [85,86]. The value creation process typically starts from a need assessment to the transformation of labor and resources to meet the aspiration of customers [44,46]. The use of social media mediates the process making firms explore hidden business opportunities in line with their business operations to make them more competitive [42,87,88]. This, perhaps, is the surest way to keep an eye on competitors to be able to tailor-make products and services towards the needs of the customers.

A key capability of social media is enabling the interconnectivity of business to business (B2B) and business to consumers (B2C). This interconnectivity fosters rapid interactions and provides low latency between SMEs and their customers, particularly on the availability of information for existing and probable customers [49]. Thus, the present study affirms the hypothesis that social media creates an opportunity for businesses to connect with their customers and make them sustainable. The findings show that social media serves as an avenue for effective interconnectivity, creates space for agility and targeted marketing, and enhances their decision-making while building loyalty [54,89,90]. This study agrees with the results from [16,38], confirming that social media has a positive influence on business connectivity. In effect, social media is better used by SMEs to exchange ideas among disparate consumers to help improve the way business is conducted [91].

SMEs' sustainability has been defined in several forms, but it is limited to the availability of money and physical resources [56,75,92]. However, a firm's sustenance in the current digital business ecosystem makes social media an integral component of the sustainability strategy. As stated by [57], SMEs need to be innovative to keep up with the changing market trends. Borah et al. [36] affirmed that leveraging social media by SMEs help improves their engagement with their customer base and creates resilience in their business models. Affirming the last hypothesis, the present study's findings demonstrate that social media helps SMEs improve their customer service and creates an opportunity

for them to enhance their stakeholder engagement and help optimize resources to meet the demand of customers, hence business sustainability [93–95]. Additionally, the study evidenced that social media usage has direct effects on achieving SMEs' sustainability in Ghana, which supports the study findings from [36].

### 6.1. Theoretical Implications

Since social media has become a common tool in the strategic direction of emerging industries, SMEs cannot be exempted. Thus, this study contributes to the ongoing discussions on the enablers and constraints of social media in transforming SMEs, particularly among emerging economies where business process improvement and automation have become common denominators. Even though the literature has been spectacular on the sustainability of firms in emerging economies [22,24], nuances of factors that promote their existence and sustainability have been under-explored. Thus, this study contributes to the literature on the growing affordances of social media in contemporary business. The study generally finds that social media offers a mammoth of opportunities for the growth of SMEs, increasing their resilience and sustainability [18,81,96,97]. In effect, social media has a positive effect on the overall activities of SMEs. The study result is important given the multiplicity of unexplored factors mitigating the growth of SMEs in developing economies [26,31]. More spectacular is the fact that it contributes to the literature on ongoing efforts at meeting the socioeconomic needs of individuals and entrepreneurs in developing economies aimed at meeting sustainable development goals through innovation and technology [36].

Again, the study findings proffer a deeper understanding of how social media provides SMEs an enabling opportunity for attaining resilience, resulting from increased profitability and performance brought forth by leveraging social media [75,92,98,99]. Given the fact that an effective social media strategy is important for SMEs to identify opportunities and explore them, the implication is that this study proffers an encompassing understanding of relevant criteria needed to enhance operational excellence.

### 6.2. Managerial Implications

The novelty in this study is arguably unmatched, given the setting and time criticality. While we acknowledged that SMEs' sustainability is an ongoing discussion, the core of the discussion has been nascent on how innovation could drive this agenda. As emerged in this study, social media positively impacts the operations of SMEs, implying that the management of SMEs must streamline their marketing mix communications in line with their social media strategy [50]. This study offers managers of SMEs a guideline on how to leverage social media to drive their marketing mix strategy. More importantly, is the fact that since extant studies have supplanted the affordances of social media in the fourth industrial revolution, understanding how innovative technologies drive this evolution is important. This will give managers of SMEs the choice of selecting sustainable innovations that will help them become customer-centric and competitive. At the core of customer satisfaction is to achieve their expectations, which also has the potential to promote brand loyalty and retention. Practically, this study offers a competitive strategy for SMEs to leverage, improve their decision-making process and become customer-centric, competitive and resilient. To be a sustainable business means the business has to be competitive. Thus, this study offers an avenue by which SMEs can be proficient in their internet and social media activities to drive their competitive edge.

### 6.3. Limitations of the Study

This study arguably is limited in terms of scope and setting. We focused on SMEs in the Ghanaian economy, for which the result may be skewed. Indeed, we acknowledge that this may have some contextual variations. Hence, the result, nonetheless, may not be encompassing. Moreover, generalizing the finding should be performed with caution also due to the sample size. We also acknowledge that using social media as a marketing tool comes with some consequences. However, this study did not explore the risk implications

of social media on SMEs. Thus, future studies could explore some of the consequences of using social media. Moreover, the study was limited to only the manufacturing industry of SMEs; future studies could explore and compare other industry-specifics to have a better view of how social media drives their sustainability agenda.

## 7. Conclusions

The research findings of this study advance the existing knowledge on social media and its usage and sustainability among SMEs from a developing country's perspective. In this day and age, social media remains a positive tool that will improve business activities, profitability and as a promotional tool for small and medium-sized businesses. However, there is limited research on the utilization of social media in achieving SMEs' sustainable performance. Therefore, this study sought to explore how social media affects the sustainability of SMEs in Ghana, given how the technology is gradually being used as a communication and marketing tool. Moreover, previous studies in this field have focused on the adoption of social media and firms' performance and user engagement. The present study investigates the extent to which the use of social media affects SMEs' manufacturing firms' sustainability. The current study hypothesized that social media has a positive impact on the operations and sustainability of SMEs. We gathered empirical data from respondents who are representatives of various SMEs in Ghana. The Partial Least Square Structural Modeling Equation (PLS-SEM) was used, particularly ADANCO 2.2.1 statistical software version, in data processing and analysis. Analysis of the data showed that, generally, social media has a positive effect on SMEs' sustainability in developing countries, specifically Ghana's SMEs' manufacturing sector. Furthermore, this study establishes that social media usage has a significant and direct effect on the profitability that creates sustainable SMEs in Ghana. In addition, this study suggests that effective social media usage empowers SMEs in developing countries to engage customers and other stakeholders. The present study offers several implications for theory and practice. The study provides an empirical understanding of how social media impacts the sustainability of businesses and offers some practical guidelines for SMEs to leverage and become sustainable. The research findings also provide a strong understanding of social media usage and its relationship with SME sustainability [36]. The study findings also advocate that social media is an effective marketing tool for SMEs in the manufacturing sector to adopt or integrate and that it has increasingly ceased to be an option for SMEs but a strategic tool to meet the increasing and sophisticated needs of the modern-day consumer in order to the improve the decision-making process and remain competitive.

**Author Contributions:** Conceptualization, E.B. and S.B.E.; Formal analysis, S.B.E. and J.A.; Funding acquisition, Z.S. and T.L.; Investigation, E.B. and H.R.; Methodology, J.A.; Project administration, Z.S. and D.Y.; Resources, E.B., H.R. and T.L.; Software, J.A.; Supervision, Z.S.; Validation, J.A. and T.L.; Visualization, S.B.E. and J.A.; Writing—original draft, E.B.; Writing—review and editing, E.B., Z.S., S.B.E. and D.Y. All authors have read and agreed to the published version of the manuscript.

**Funding:** This research received external funding from Doctoral Research Innovation Fund Project of CWAS of UESTC accredited by Ministry of Education, China. Project No (CXJJ2021122504).

**Institutional Review Board Statement:** Not applicable.

**Informed Consent Statement:** Not applicable.

**Data Availability Statement:** This submission has no linked research data sets. The following reason is given: It is basic survey research and data will be made available upon request.

**Conflicts of Interest:** The authors declare that they have no financial or personal relationships which may have inappropriately influenced them in writing this article.

## Appendix A

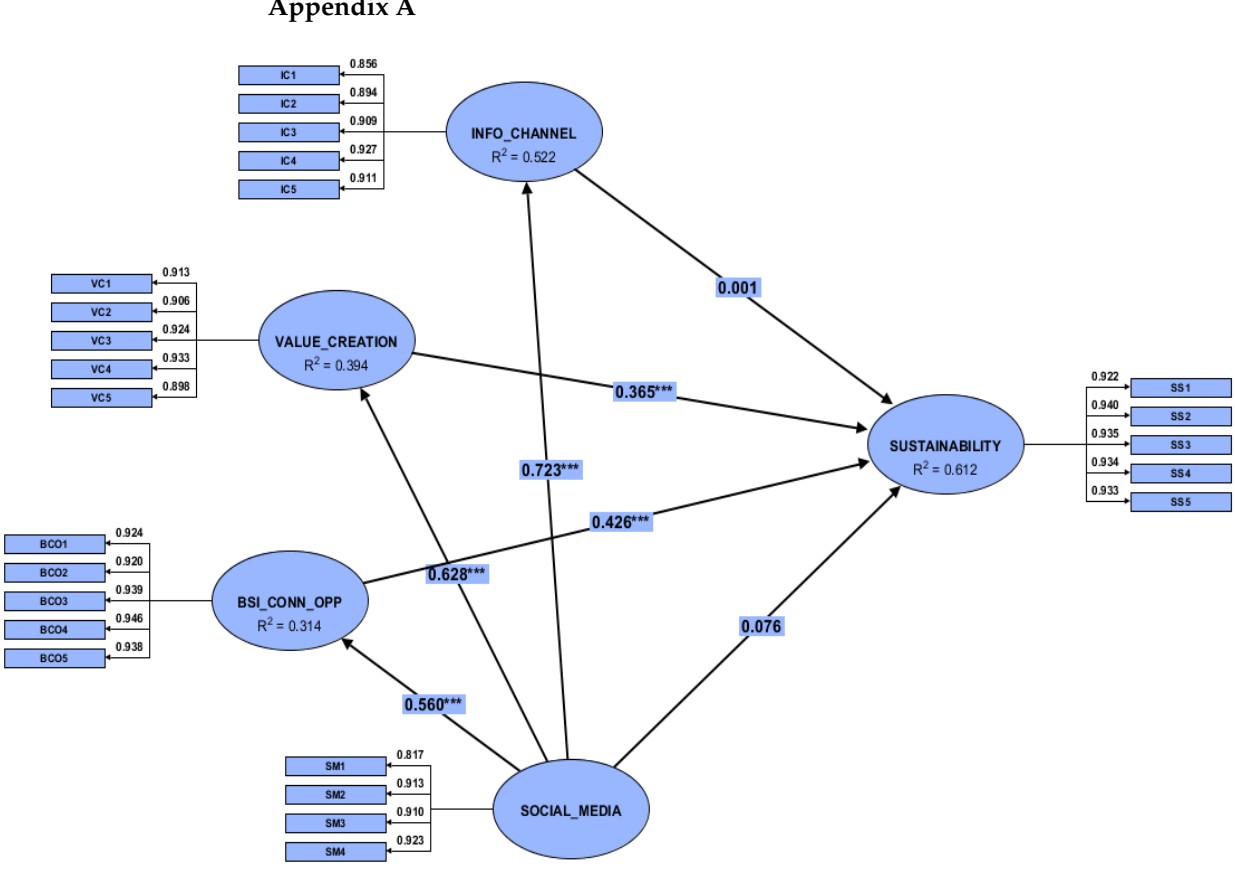

**Figure A1.** Empirically Tested Research Model (*** $p < 0.001$).

## Appendix B. Article Questionnaire and Measurements

| | Variable Measures |
|---|---|
| **Social media** | **SM1**: Social Media are user-friendly.<br>**SM2**: Social media helps SMEs to engage in business with distance customers<br>**SM3**: Create meaningful relationship with customers<br>**SM4**: Helps in creating and increasing brand awareness |
| **Information Channel** | **IC1**: Social media can be used as a means to engage or interact with customers<br>**IC2**: Social media gives customers adequate information on a particular product or service before it is purchased or consumed.<br>**IC3**: Social media can be used to attract customers and reach the masses<br>**IC4**: Social media serve as a strategic informational channel for SMEs<br>**IC5**: SMEs use social media as a channel to obtain feedback from customers |
| **Value Creation** | **VC1**: Firms can use social media to improve customer relationships and increase market share.<br>**VC2**: Social media help firms to know more about customers' perceptions of the company<br>**VC3**: Social media lower marketing campaign costs, and helps in awareness creation, creating a meaningful customer-driven product innovation.<br>**VC4**: Social media usage helps in customer brand loyalty and trust |
| **Business Connection and Opportunities** | **BCP1**: Social media facilitate business connections with their customers and other stakeholders via disseminating information at a reduced cost.<br>**BCP2**: SMEs get access to a larger market, thereby expanding the customer base both locally and globally.<br>**BCP3**: Social media platforms directly target customers with marketing campaigns thus promoting new products or services and building brand awareness.<br>**BCP4**: Social media is a strategic online platform for job searching, recruitment, and career growth for SMEs<br>**BCP5**: Social media helps SMEs to compete with large companies in the area of marketing for competitive advantage. |

| | |
|---|---|
| **Sustainability** | **SS1**: Social media is a valuable tool for SMEs' marketing<br>**SS2**: In my view, social media enhances the productivity of the firm<br>**SS3**: Customers are adequately compatible in using social media to patronize the firms' products and services.<br>**SS4**: In my view, social media makes it possible to identify customer demands and satisfy them accordingly<br>**SS5**: Our firm is compatible with using social media for marketing purposes. |

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
