# Peer review of "Social Media Usage and SME Firms’ Sustainability: An Introspective Analysis from Ghana"

_sustainability, doi:10.3390/su14159433_

Round 1

Reviewer 1 Report

I believe it is an interesting manuscript that may attract our target readers. however, some limitations are still existed. 

1, the process of conducting questionaire should be detail described. i wonder the different people in same organization may provide very different insights. 

2, lacking of lastest studies in part of literature review pertaining to social media in marketing area.

3, each tables should be three-line table format.

Reviewer 2 Report

The biggest issue of this work is the measurement of 'sustainability' along with other variables, for instance, 'Information Channel (IC), Value Creation (VC), Busi-ness Connections and Opportunities (BCO)'. There is no clear explanation on how these variables/indicators were measured.

Meanwhile, the design of the questionnaire is not clear and the no. of respondents is relatively small comparing to the total no. of SME in Ghana. 

Round 2

Reviewer 2 Report

Thank you for your response and explanations. The current version looks good.